# Crystal Structure, Photophysical Properties and Antibacterial Activity of a Cd(II) Complex with *Trans*-2,3,4-Trimethoxycinnamic Acid and 4,4′-Bipyridine Ligands

**DOI:** 10.3390/polym16182643

**Published:** 2024-09-19

**Authors:** Linyu Wang, Xiao Han, Qun Liu, Jianye Li, Zhifang He

**Affiliations:** 1Hebei Key Laboratory of Heterocyclic Compounds, Hebei Center for New Inorganic Optoelectronic Nanomaterial Research, College of Chemical Engineering & Material, Handan University, Handan 056005, China; wanglinyu.abc@163.com (L.W.); jianyeli1988@163.com (J.L.);; 2Tianjin Institute for Food Safety Inspection Technology, Tianjin 300308, China

**Keywords:** Cd(II) complex, photophysical properties, bacteriostatic activity

## Abstract

A new coordination polymer {[Cd(C_12_H_13_O_5_)_2_(4,4′-bpy)(H_2_O)_2_]}_n_ (**Cd-Tmca-bpy**) was constructed with trans-2,3,4-Trimethoxycinnamic acid (HTmca) and 4,4′-Bipyridine (4,4′-bpy) ligands. This complex was structurally characterized on the basis of elemental analysis, infrared (IR) spectroscopy, powder X-ray diffraction and thermogravimetric analyses. X-ray crystallography revealed that the complex was monoclinic, space group C2/c. The Cd(II) ion in the complex was six coordinated, adopting an octahedron geometry. The neighboring Cd(II) ions linked linear ligand 4,4′-bpy molecules to form an infinite 1D chain. The 1D chain was further interlinked by O–H···O and C–H···O hydrogen bonds, resulting in a 3-D supramolecular framework. Meanwhile, the photoluminescence spectrum of the Cd(II) complex at room temperature exhibited an emission maximum at 475 nm. Using the time-dependent density functional theory (TD-DFT) method, the electronic absorption spectra of the Cd(II) complex was predicted. A good agreement was achieved between the predicted spectra and the experimental data. Bioactivity studies showed that the complex exhibited significant inhibition halos against *Pseudomonas aeruginosa* (*P. aeruginosa*) and *Staphylococcus aureus* (*S. aureus*).

## 1. Introduction

Coordination complexes, a novel kind of porous hybrid materials created through self assembly of metal ions (clusters) and organic ligands, have drawn extensive attention and sparked intense discussions. This is not only because of their spatial orientations, near-infinite number of possible combinations of metal–ligand, easily achievable synthesis and characterization, and low cost, but also due to their broad and promising application prospects in diverse fields such as catalytic, medical, magnetic and optical materials [1,2,3,4,5,6,7,8]. Infections triggered by pathogenic microorganisms including bacteria, fungi and viruses have long been a threat to human health around the world. In recent years, excessive use of traditional antibiotics has led to emergence of multidrug-resistant (MDR) bacteria, presenting a substantial threat to public health [9,10]. The multidrug-resistant (MDR) bacteria are particularly troublesome. They encompass Gram-negative bacteria such as *Acinetobacter baumannii* (*A. baumannii*), *Klebsiella pneumoniae* (*K. pneumoniae*), *Escherichia coli* (*E. coli*), *Pseudomonas aeruginosa* (*P. aeruginosa*) and *Serratia marcescens* (*S. marcescens*), as well as Gram-positive bacteria such as *Streptococcus pneumonia* (*S. pneumonia*), *Staphylococcus aureus* (*S. aureus*) and *Staphylococcus epidermidis* (*S. epidermidis*) [11].

Over the past ten years, development of antimicrobial agents that can inhibit bacterial growth, prevent biofilm formation and achieve sterilization has become a focal point in antibacterial applications. The coordination complex, a new kind of high-performance material, holds potential for antimicrobial use.

Coordination complexes, which can serve as outstanding metal-ion reservoirs for release of metal ions, have emerged as highly promising materials for a wide range of antimicrobial applications. In comparison with traditional antibacterial agents, coordination complex antibacterial agents owe their merits to different functions. Coordination complexes composed of bactericidal metal ions (Ag^+^, Zn^2+^, Co^2+^ and Cu^2+^) and certain organic antimicrobials/natural bioantimicrobials (imidazoles and porphyrins) have been constructed. Release of components with bactericidal activity via controlled/stimulated decomposition can be initiated by a change in biological environment, such as reduction in pH or laser irradiation [12,13,14,15]. 

Moreover, the type of ligand and the configuration of the complexes play crucial roles in the antimicrobial properties of coordination complexes. A complex can serve as a reservoir for various antimicrobial agents that can be used in detection of harmful pathogens and destruction of these pathogens. Coordination complexes, which feature high porosity, large specific surface areas, tunable structures and excellent physicochemical stability, can facilitate efficient encapsulation or loading of other materials into their pores. Furthermore, abundant surface active groups can allow for modification of other materials on their surfaces. This characteristic holds great significance. The presence of these numerous surface active groups provides a platform for attachment and integration of various other materials. When other materials are modified on their surfaces, it paves the way for creation of innovative combinations. This is extremely beneficial for obtaining antimicrobial composites with dual effects, enhancing functionality and effectiveness in combating harmful microorganisms [16].

The self assembly of coordination complexes is accomplished through providing a certain quantity of custom-designed organic/inorganic building blocks for assembly by means of directional interactions such as metal coordination. Combining different organic ligands and metal ions/metal clusters has proven to be a very reliable strategy for synthesizing ideal functions. Most coordination-complex architectures are purposefully designed based on size, shape, and surface properties of the building blocks, as well as chemical environments. Occasionally, unexpected structures and functions have been encountered due to more intricate self-assembly processes than anticipated. Fascinatingly, due to the special electron structure and the metal–metal bond, many structural characterizations of representative *d*^7–10^ transition metal complexes show excellent luminescence, especially for *d*^6^, *d*^8^ and *d*^10^ transition metal complexes. In general, coordination polymers containing a *d*^10^ configuration metal ion (such as Zn^2+^ or Cd^2+^) have been observed to display photoluminescent properties [17,18,19]. For Cd(II) complexes, due to the *d*^10^ electron configuration and large ionic radius of Cd(II) ions (meaning easy to compress), the Cd(II) ion exhibits different coordination modes and leads to formation of diverse structure designs. 

Cinnamic acid, a remarkable organic acid, can be obtained from cinnamon bark or benzoin. Its unique structure, which consists of a benzene ring, an alkene double bond and an acrylic acid functional group, endows it with great potential for modification. Through various compounds, the functionalities can be altered and enhanced. This enables production of bioactive agents with even more potent effects. Notably, the role of cinnamic acid derivatives in different areas has been reported. In treatment of cancer, the derivatives may exhibit properties that inhibit tumor growth. For bacterial infections, they can combat pathogens effectively. In diabetes management, they might help regulate blood sugar levels. In neurological diseases, they could potentially offer therapeutic benefits, showing promise in multiple aspects of healthcare [20,21,22]. However, some metal complexes with cinnamic acid derivatives remain to be further studied [23,24,25]. The antibacterial activity of complexes may be more effective when compared to the cinnamic acid derivatives, thus can be regarded as promising therapeutic agents.

In this study, we designed and synthesized a Cd(II) complex {[Cd(C_12_H_13_O_5_)_2_(4,4′-bpy)(H_2_O)_2_]}_n_ (**Cd-Tmca-bpy**) based on cinnamic acid derivatives. The complex was characterized by elemental analysis, infrared spectroscopy and ultraviolet spectroscopy, and the photophysical properties and antibacterial activities of the ligands and complexes were studied. X-ray crystallography revealed that the complex was monoclinic, space group C2/c. The Cd1 was six coordinated with an octahedron geometry, and the neighboring Cd^2+^ ions linked linear ligand 4,4′-bpy molecules forming an infinite 1D chain and were further interlinked through O–H···O hydrogen bonds to form a 2D layer. The layers were further connected by three C–H···O hydrogen bonds to generate a 3D supramolecular framework. Meanwhile, the photophysical properties of the Cd(II) complex have also been investigated in detail. The photoluminescence spectrum of the Cd(II) complex at room temperature exhibited an emission maximum at 475 nm. Optimized conformation of the Cd(II) complex has been calculated with density functional theory (DFT). By using the time-dependent DFT(TD-DFT) method, electronic absorption spectra of the Cd(II) complex have been predicted and a good agreement can be achieved with the experimental data. The bioactivity studies showed that the complex exhibited significant inhibition halos against *Staphylococcus aureus* and *Pseudomonas aeruginosa*.

## 2. Experimental

### 2.1. Materials and Physicochemical Measurements 

*Trans*-2,3,4-Trimethoxycinnamic acid (HTmca) and 4,4′-Bipyridine (4,4′-bpy) of 99% purity have been used without further purification. The Fourier Transform Infrared (FT-IR) spectra were measured using a Bruker TENSOR 27 FT-IR spectrophotometer. The measurement range spanned from 4000 to 400 cm^−1^. A resolution of 2 cm^−1^ was employed for the analysis. The determination was carried out via the conventional KBr pellets method. This method involves preparing pellets by mixing the sample with potassium bromide. The spectrophotometer then analyzes the infrared absorption of the sample in this specific range, providing valuable information about the chemical bonds and functional groups present in the sample. A Vario EL cube CHNS elemental analyzer is used to determine carbon, hydrogen and nitrogen content in coordination complex. Powder X-ray diffraction (PXRD) patterns of the crystal structure were collected using a Bruker AXS D8 Advance X-ray diffractometer with Cu Kα radiation (λ  =  1.5406 Å) at 40 kV and a scanning scope of 5–60° (2*θ*). The sample was spread as a thin layer on the circular recess of an XRD sample holder. Thermogravimetric analyses (TGA) were carried out under a nitrogen atmosphere. A TC/DTA7200 thermogravimetric analyzer was utilized for this purpose. The heating rate was set at 10 °C/min. These analyses help in determining the thermal stability and decomposition behavior of the sample. On the other hand, UV/Vis absorption spectra were obtained using a T6 UV-Vis Spectrophotometer. These spectra provide information about the absorption of ultraviolet and visible light by the sample. Additionally, photoluminescence measurements were performed at room temperature on a Hitachi F-7000 spectrometer. This allows for the study of the emission of light by the sample when excited by an appropriate source.

### 2.2. Single-Crystal Structure Determination

A suitable single crystal of **Cd-Tmca-bpy** was carefully picked under an optical microscope. This single crystal was then mounted on a glass fiber. For collecting single-crystal X-ray diffraction intensity data, a Bruker APEX-II CCD diffractometer with Cu Kα radiation (λ = 1.5478 Å) was employed. During the data collection process, the crystal was maintained at a temperature of T = 298 K. Regarding the structure, software such as Olex2-1.2, which is used to prepare material for publication, was utilized [26,27]. The structure solution and refinement were carried out with the ShelXT structure solution program using Intrinsic Phasing. Subsequently, the refinement was performed with the ShelXL refinement package using Least Squares minimization. Figures were created with Diamond. Crystallographic data and structure processing parameters are neatly summarized in Table 1. Selected bond lengths and angles for the complex are given in Table 2 and Table 3. The hydrogen bond of complex are presented in Table 4. CCDC: 2264022.

### 2.3. Synthesis of Complex **Cd-Tmca-Bpy**

A mixture containing HTmca (0.0238 g, 0.1 mmol), 4,4′-bpy (0.0156 g, 0.1 mmol) and Cd (CH_3_COO)_2_·2H_2_O (0.0266 g, 0.1 mmol) was dissolved in 5 mL of H_2_O and CH_3_CH_2_OH (4:1). The mixture was placed in a 10 mL glass bottle. Subsequently, it was heated to 100 °C. At this temperature, it was maintained for three days. After that, it was removed from the constant temperature oven. Then it was cooled to room temperature at a rate of 5 °C·h^−1^. Colorless crystals of **Cd-Tmca-bpy** were obtained. These crystals were washed several times with CH_3_CH_2_OH and finally dried. (Yield: 63%). Elemental analysis for C_34_H_38_CdN_2_O_12_, calcd (%): C, 52.41; H, 4.92; N, 3.60. Found (%): C, 49.80; H, 4.65; N, 3.44.

### 2.4. Computational Details

All calculations are performed in the Gaussian 09 package [28,29]. The structures of the ground state (S0) and the first excited state (S1) were fully optimized by using the B3LYP functional and 6-31+G(d) basis set in DFT and TD-DFT methods [30,31,32,33,34,35]. The solvation effect was evaluated by using a conductor-like polarized continuum model (C-PCM) [36,37,38,39]. 

### 2.5. Agar Well Diffusion Method

The antibacterial activity of the complex against nine microorganisms was evaluated by using the agar well-diffusion method, namely against *Staphylococcus epidermidis* (14990), *Bacillus subtilis subsp. Spizizenii* (6633), *Escherichia coli* (25922), *Staphylococcus aureus* (12600), *Pseudomonas aeruginosa* (15692), *Streptococcus mutans* (25175), *Moraxella catarrhalis* (25238), *Pseudomonas fluorescens* (13525), and *Enterococcus hirae* (8043).

Petri dishes containing 25 mL of solid media were prepared. The test pathogenic strains were suspended in sterile water at an OD_580_ value of about 1.0 (approximate 1 × 10^7^ CFU/mL). Then 200 μL microbial suspensions were spread across the agar surface of Petri dishes using a cotton wool swab. Wells were aseptically punched into the agar using a sterile pipette tip. The test substance (20 μL, dissolved in DMSO at a concentration of 2 mg/mL) was added into the wells and then incubated at suitable temperature for 24 h. DMSO was used as negative control. The antimicrobial activity was evaluated by measuring the diameter of the zone of inhibition. The diameter of each well was 3 mm.

## 3. Results and Discussion

### 3.1. Crystal Structure of {[Cd(C_12_H_13_O_5_)_2_(4,4′-bpy)(H_2_O)_2_]}_n_ (Cd-Tmca-bpy)

The complex **Cd-Tmca-bpy** crystallized in the monoclinic space group C2/c, with a = 28.753(12), b = 11.848(5), c = 10.632(5) Å, α = 90, β = 105.513(13), γ =90°, V = 3490(3) Å^3^, Z = 4. The asymmetric unit contained half Cd(II) ions, one Tmca^1−^ anion, half 4,4′-bpy and one coordination water molecule. As is shown in Figure 1, Cd1 was six coordinated with an octahedron geometry, completed by two oxygen atoms from different Tmca^1−^ ligands, two nitrogen atoms originating from two distinct 4,4′-bpy ligands and two oxygen atoms from coordinated H_2_O molecules. The average distances of the Cd–O bond were 2.296 and 2.297 Å and were consistent with the previously reported literatures [40]. The carboxylate groups of the Tmca^1−^ anion around the Cd1 ion adopted the unidentate coordination mode. The dihedral angle between the two phenyl rings of the 4,4′-bpy molecules was 48.869°.

The adjacent Cd ions connected the linear ligand 4,4′-bpy molecules to form an endless 1D chain parallel to the c direction (Figure 2). All Tmca^1−^ ligands were suspended at both sides of the chain, and the dihedral angle between the two phenyl rings of the distinct Tmca^1−^ ligands was 42.651°.

There were two O–H···O hydrogen bonds and three C–H···O hydrogen bonds (Table 4). Each coordinated water molecule in **Cd-Tmca-bpy** was connected with carboxylate groups (O1-C1-O2) by the hydrogen bonds O6-H6A···O1[x, −y + 1, z − 1/2] and O6-H6B···O2 (Figure 3). The hydrogen bond O6-H6A···O1[x, −y + 1, z − 1/2] had a bond distance d (O6···O1[x, −y + 1, z − 1/2]) of 2.717 Å and a bond angle ∠O6-H6A···O1[x, −y + 1, z − 1/2] of 148.54°. Meanwhile, the hydrogen bond O6-H6B···O2 had a bond distance d_(O6···O2)_ of 2.613 Å and a bond angle ∠O6-H6B···O2 of 147.88°. The 1D chains were further interlinked through O–H···O hydrogen bonds to form a 2D layer (Figure 3). The layers were further joined together by three C–H···O hydrogen bonds to form a 3D supramolecular framework (Figure 4).

### 3.2. PXRD

The XRD spectrum in Figure 5 indicates a high crystallinity of the coordination structure, and the phase purity of the products were further affirmed by XRD analyses. The experimental patterns were remarkably similar to the simulated patterns that were based on the single-crystal data. This close resemblance serves as strong confirmation that **Cd-Tmca-bpy** had been successfully obtained as a pure crystalline phase [41].

### 3.3. FT-IR Spectra

The FT-IR spectra of HTmca, 4,4′-bpy and the complex **Cd-Tmca-bpy** showcased several distinct bands that emerged within the spectral region spanning from 500 to 4000 cm^−1^. As presented in Figure 6, these spectra offer a wealth of information. The existence of these bands is highly significant as it provides crucial insights into the chemical bonds and functional groups that were present within these substances. Meticulous analysis of these spectra can delve deep into the molecular structure and properties of HTmca, 4,4′-bpy and the complex **Cd-Tmca-bpy**. This analysis can help in understanding the nature of the interactions between the different atoms and groups within these compounds, as well as their potential applications in various fields. The wide adsorption peaks at about 3550, 3480 and 3415 cm^−1^ are the characteristic peaks of -OH groups. As shown in Figure 6, there was no absorption near 1700 cm^−1^, indicating that all the carboxylic acid groups in the organic ligand had removed protons. It is important to note that the separations (denoted as Δ) between the asymmetric stretching vibration frequency of the carboxylate group υ_a_ (COO) and the symmetric stretching vibration frequency of the carboxylate group υ_s_ (COO) vary depending on the nature of complexes. Specifically, these separations are different for unidentate complexes, chelating (bidentate) complexes and bridging complexes. This variation in separations can be utilized to determine the mode of coordination of the carboxylate groups in a given complex [42]. In **Cd-Tmca-bpy**, the carboxylate groups exhibited υ_a_ (COO) and υ_s_(COO) at 1639 and 1365 cm^−1^ (Δ = 274 cm^−1^). This Δ value is comparable to those of carboxylate groups in unidentate coordination mode. The absorption peak from the stretching C=N bands of **Cd-Tmca-bpy** (1604 cm^−1^) exhibited a blue shift compared with the characteristic peak of 4,4 ‘-bpy stretching at 1595 cm^−1^, which was due to the coordination interactions between the metal ions and the pyridine nitrogen [43,44]. Based on the above results, they are consistent with the crystallographic details deduced from the refinement.

### 3.4. Thermogravimetric Analyses

The crystal structure and thermal properties of **Cd-Tmca-bpy** were studied by thermogravimetric analyses in a nitrogen atmosphere with a heating rate of 10 K min^−1^ at T = 50∼1000 °C. As shown in Figure 7, the thermal decomposition processes showed a three-step weight-loss stage. Up to 165 °C, the first step showed a mass loss (∆m_exp_ = 4.63%) suggesting release of coordinated H_2_O molecules (∆m_calc_ = 4.62%). In the temperature range of 220–555 °C, the total weight loss was 78.13%, corresponding to combustion of two Tmca^1−^ ligands and one 4,4′-bpy ligands. Further heating to 600 °C led to decomposition of the whole framework. After the reactions, there was a residual amount of unreacted substance left in the crucible (17.23%). This corresponded to the ratio of the Cd and O components, indicating that the final product was CdO (Calcd: 16.48%). By combining thermogravimetric analyses with elemental analysis, it is speculated that there may be a small amount of cadmium compound mixed in the sample. This leads to a residual amount higher than the theoretical value of CdO produced only by thermal decomposition of the complex.

It was found that the decomposition temperature of the complex was 220 °C, which indicates the complex was quite stable to heat. It is widely acknowledged that the construction and properties of coordination compounds can be potentially influenced by a variety of inter/intra-molecular interactions. Among these interactions are C–H···π interactions, where a hydrogen atom from a C–H bond interacts with the π-electron cloud of an aromatic ring. Additionally, there are N–O···π interactions, which involve the oxygen atom from a nitro group or similar moiety interacting with the π-electron cloud. Hydrogen bonding, a crucial interaction, occurs between a hydrogen atom bonded to an electronegative atom and another electronegative atom. Furthermore, π–π interactions, which are stacking interactions between aromatic rings, also play a significant role. These diverse interactions can have a profound impact on structural stability, electronic properties and other characteristics of coordination compounds [45]. The result of the TG analyses is basically in accordance with that of the structure determination of **Cd-Tmca-bpy**. There were two O–H···O hydrogen bonds and three C–H···O hydrogen bonds in **Cd-Tmca-bpy**.

### 3.5. Solid-State Luminescence Emission

Coordination complexes are generally constructed by metal nodes (metal clusters or ions) and organic ligands, periodically. Selection of central metal nodes (metal clusters or ions) and organic ligands are of great importance for the fabrication of coordination complexes with excellent optical properties. The *d*^10^ metal ions and conjugated organic ligands are good choices for excellent fluorescence performance. The complex **Cd-Tmca-bpy** contained *d*^10^ metal ions Cd(II) and conjugated π-bond aromatic carboxylic acid ligands, and thus it can be viewed as a new type of fluorescent complex. Solid-state luminescence of HTmca, 4,4′-bpy and the complex **Cd-Tmca-bpy** were investigated at room temperature (Figure 8).

Complex **Cd-Tmca-bpy** emitted a faint blue color, with a maximum emission at 475 nm when excited at 330 nm. The fluorescent emission band of the HTmca ligands occurs at λ_em_ = 438 nm under λ_ex_ = 330 nm. This emission can be attributed to the π*→π or π*→n electronic transitions. The Cd(II) ion, with its *d*^10^ electronic configuration, is difficult to oxidize or reduce. Furthermore, the free 4,4′-bpy ligand does not have any emission in the range between 400 nm and 800 nm [46,47]. As a result, the emissions of **Cd-Tmca-bpy** cannot be ascribed to either ligand-to-metal charge transfer bands (LMCT) or metal-to-ligand charge transfer bands (MLCT). Instead, they might be attributed to fluorescent emissions originating from an intraligand (π-π*) excited state [48,49,50]. Comparing with HTmca and 4,4′-bpy, the emission of **Cd-Tmca-bpy** were pronouncedly red shifted (53 nm and 37 nm, respectively). The significant red shift of the emission that occurred in the title complex implies that the ligands coordinated in a unidentate coordination mode to the metal ions, which effectively increased the rigidity of the ligands. Moreover, the structural reason that **Cd-Tmca-bpy** possessed a largely red-shifted emission can also be assigned to its 3D supramolecular framework, which was based on two O–H···O hydrogen bonds and three C–H···O hydrogen bonds.

### 3.6. Solvent Effect

The solid-state fluorescence spectra showed that the complex had good fluorescence properties. The fluorescence-sensing properties of **Cd-Tmca-bpy** in solution were investigated. Each solvent has its own unique properties that can influence emission behavior of compounds. As shown in Figure 9, the spectra showed a clear blue shift with the different polarity of solvents. It is noteworthy that the relative fluorescence intensity of **Cd-Tmca-bpy** was quenched when DMF and DMAC were used as dispersants, while the relative fluorescence intensity of **Cd-Tmca-bpy** was strongest in ethanol and acetonitrile. Analyzing these emission spectra in different solvents can provide insights into the solvent-dependent behavior of compound **Cd-Tmca-bpy**. This information can be useful for understanding its photophysical properties and potential applications in different environments. Therefore, ethanol was selected as the dispersant of **Cd-Tmca-bpy**, which could be used for the performance of fluorescence sensing.

### 3.7. UV–Visible Absorption Spectra and TD-DFT Calculations

Single-crystal X-ray diffraction (SCXRD) is a potent non-destructive approach that enables unambiguous identification of crystalline phases, determination of crystal structure (including unit cell parameters, space group, atomic coordinates and atomic occupancies) and, if necessary, phase composition. In this study, the structures and compositions of the complex **Cd-Tmca-bpy** were analyzed using single-crystal X-ray diffraction. We carried out calculations based on the structure from SCXRD with the aid of the crystallographic information file (CIF). The structure of **Cd-Tmca-bpy** is described above and is compared with B3LYP-optimized geometries of the complex in Figure 10. Simultaneously, from TD-DFT calculations on complex **Cd-Tmca-bpy**, we acquired the computational UV–Vis spectrum and utilized it to compare with the experimental UV–Vis spectrum. The experimental UV/Vis values are in excellent agreement with the theoretical optical transitions values.

In the case of the complex **Cd-Tmca-bpy**, HOMO, HOMO–1, LUMO+2 and LUMO+3 orbitals primarily derived from the HTmca ligand, whereas the electron density on LUMO and LUMO+1 orbitals were mainly localized on the d-orbitals of the 4,4′-bpy ligand and the transition metal. It is clearly evident that the energy gap between HOMO and LUMO for compound **Cd-Tmca-bpy** was 4.3935 eV (Figure 11) [51]. This energy gap played a crucial role in determining the electronic properties and reactivity of the compound. A relatively large energy gap indicates a higher stability and lower reactivity, while a smaller gap may suggest greater ease of electron transfer and potentially higher reactivity. Understanding the energy gap is important for predicting and analyzing the behavior of the compound in various chemical and physical processes.

The UV–Vis absorption spectra of complex **Cd-Tmca-bpy** were investigated at room temperature in various solvent systems. Studying these UV–Vis absorption spectra in different solvents provides a better understanding of the compound’s electronic structure and its response to different solvent environments. This analysis provides valuable information about the electronic structure and optical properties of the complex. Different solvents can have significant impacts on absorption spectra due to differences in solvent polarity, hydrogen bonding capabilities and other properties. Studying the complex in different solvent systems provided insights into the solvent-dependent behavior of the complex and its interaction with the surrounding environment. This knowledge can be useful for understanding the photophysical and photochemical properties of the complex and for potential applications in areas such as optoelectronics and photocatalysis. The complex **Cd-Tmca-bpy** showed an absorption maximum at 293 nm in ethanol, and the corresponding calculated absorption band was located at 291 nm. Accordingly, this band can be attributed to the transitions of HOMO to LUMO+3 with a contribution of 40% and HOMO-1 to LUMO+2 with a contribution of 30%. The corresponding value of oscillator strength was at 1.5541 atomic units. This transition may be designated as a π–π* transition. This indicates that the electronic excitation responsible for this absorption band involved the transfer of an electron from a π orbital to a higher energy π* orbital. The specific contributions from the different orbital transitions suggested a complex electronic structure for the compound. The oscillator strength provided a measure of the intensity of the transition, with a higher value indicating a stronger absorption. Understanding these electronic transitions is important for understanding the optical and electronic properties of the compound [52,53].

### 3.8. Antibacterial Activity

The complex **Cd-Tmca-bpy** underwent preliminary antimicrobial screening by means of well-diffusion assays against a diverse range of organisms, including five Gram-positive bacteria and four Gram-negative bacteria. This screening process was crucial for determining the potential antimicrobial activity of the complex. Well-diffusion assays involve placing a sample of complex in a well on an agar plate inoculated with the test organisms. Diffusion of the complex from the well into the surrounding agar allows for assessment of its ability to inhibit growth of the bacteria. By testing against both Gram-positive and Gram-negative bacteria, a more comprehensive understanding of the complex’s antimicrobial spectrum can be obtained. This initial screening can provide valuable insights into potential applications of the complex in the field of antimicrobial therapy [54]. The complex **Cd-Tmca-bpy** displayed activity in these assays against *Staphylococcus epidermidis* (14990), *Bacillus subtilis subsp. Spizizenii* (6633) and *Escherichia coli* (25922), while it showed significant inhibition halos against *Staphylococcus aureus* (12600) and *Pseudomonas aeruginosa* (15692), as shown in Table 5. However, there was no antibacterial activity against the other four species, *Streptococcus mutans* (25175), *Moraxella catarrhalis* (25238), *Pseudomonas fluorescens* (13525) and *Enterococcus hirae* (8043).

## 4. Conclusions

One novel Cd(II) coordination polymer has been synthesized, utilizing trans-2,3,4-Trimethoxycinnamic acid accompanied by 4,4′-Bipyridine. The central Cd(II) ion was octahedrally coordinated in a N_2_O_4_ donor set, forming an infinite 1D chain. The hydrogen bonding interactions further connected the chains to generate a 3D supramolecular architecture. Hydrogen bonding interactions assembled the 1D chains into 3D networks and further stabilized the 3D framework structure of complex. The thermal decomposition processes of the complex were divided into three steps. The emission spectrum of Cd(II) complex at room temperature exhibited an emission maximum at 475 nm. In order to understand the electronic transitions and the spectral character of complex **Cd-Tmca-bpy** in different solvents, the optimal geometric structure of complex **Cd-Tmca-bpy** was determined by DFT calculation, and the HOMO and LUMO energies were measured by the time-dependent TD-DFT approach. The bacteriostatic activity of the complex **Cd-Tmca-bpy** against *Staphylococcus aureus* and *Pseudomonas aeruginosa* showed obvious effects.

## Figures and Tables

**Figure 1 polymers-16-02643-f001:**
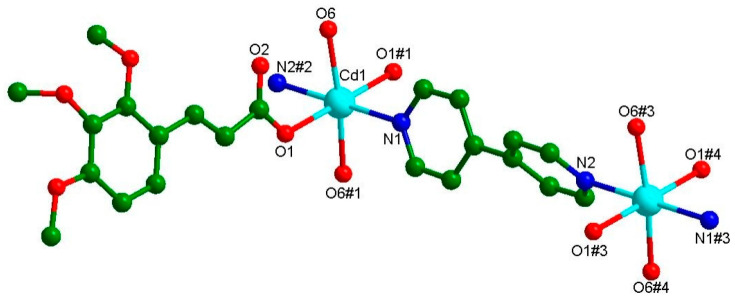
The coordination environment of Cd(II) with the atom numbering scheme presents a detailed view of the arrangement of atoms around the Cd(II) ion.

**Figure 2 polymers-16-02643-f002:**
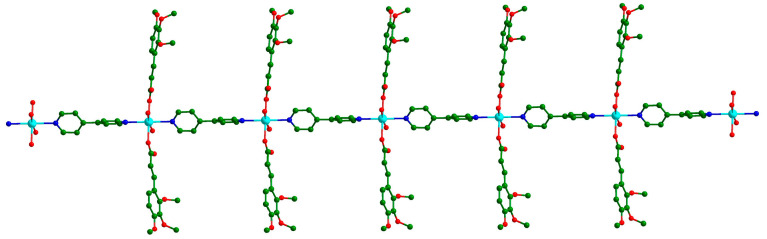
In complex **Cd-Tmca-bpy**, the one-dimensional chain extends along the *c* axis.

**Figure 3 polymers-16-02643-f003:**
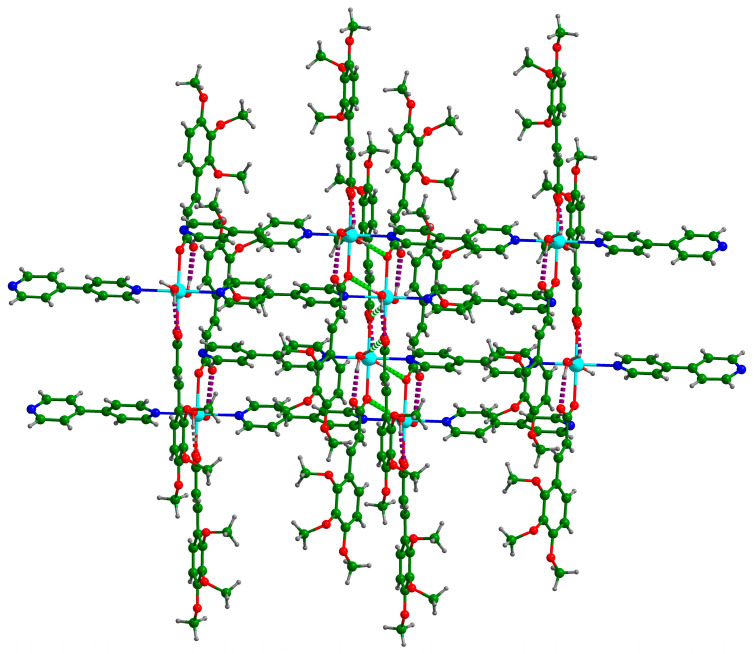
The two-dimensional layer structure in the complex is formed by the establishment of hydrogen bonds between the chains. The presence of these hydrogen bonds is indicated by the dotted lines.

**Figure 4 polymers-16-02643-f004:**
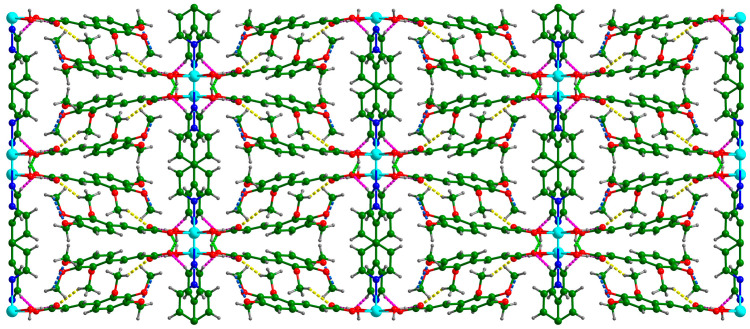
Supramolecular networks in the system are constructed through the formation of hydrogen bonds between the chains. The dotted lines serve as visual indicators of these hydrogen bonds.

**Figure 5 polymers-16-02643-f005:**
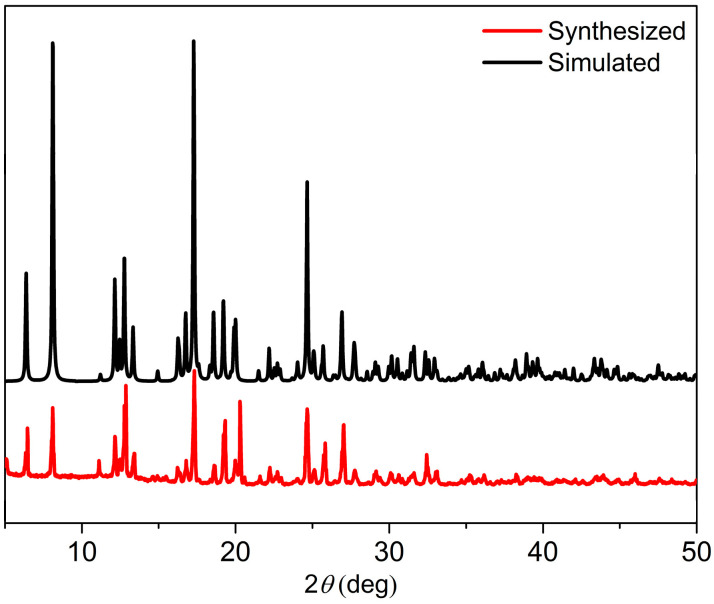
The simulated and experimental PXRD patterns of **Cd-Tmca-bpy**.

**Figure 6 polymers-16-02643-f006:**
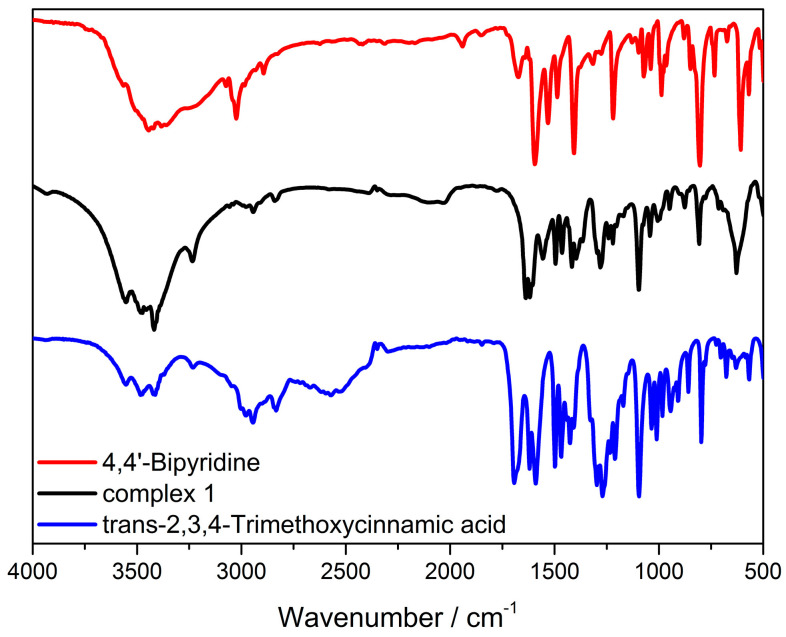
The FT-IR spectra of HTmca, 4,4′−bpy and complex **Cd-Tmca-bpy**.

**Figure 7 polymers-16-02643-f007:**
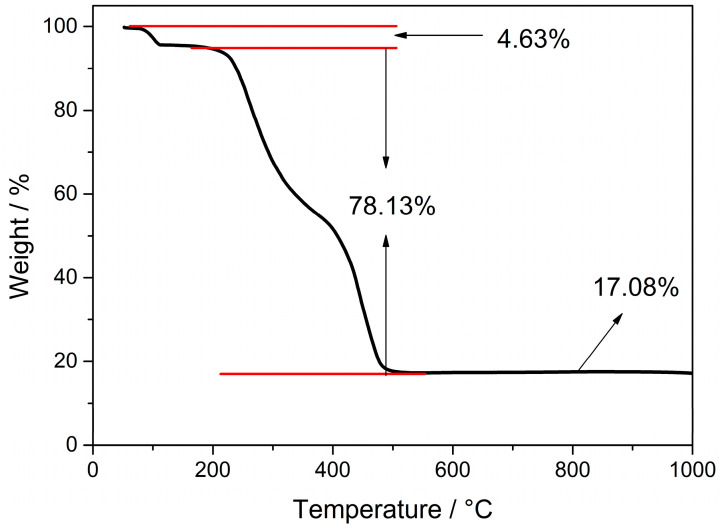
Thermogravimetric analysis curve of **Cd-Tmca-bpy**.

**Figure 8 polymers-16-02643-f008:**
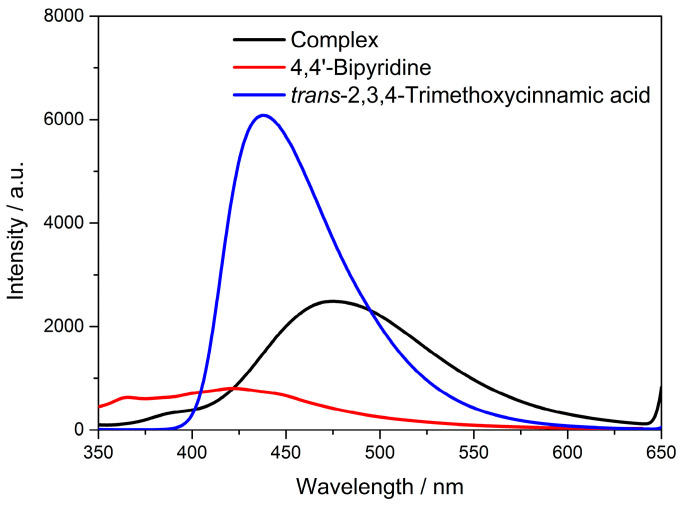
The emission spectra of HTmca, 4,4′-bpy and complex **Cd-Tmca-bpy**.

**Figure 9 polymers-16-02643-f009:**
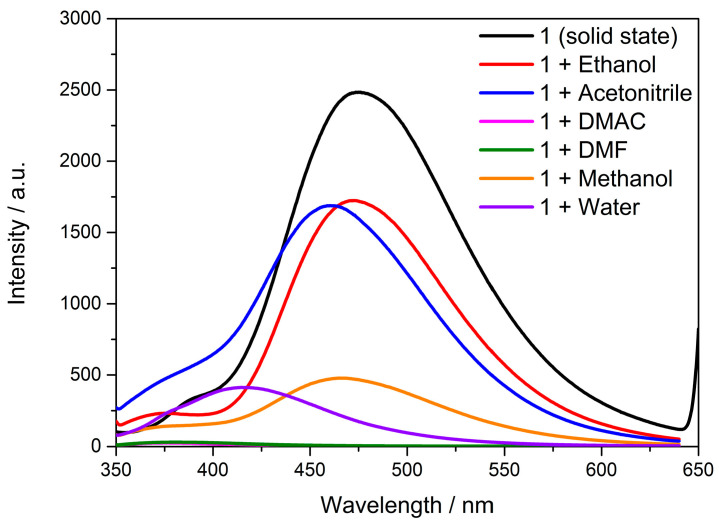
The emission spectra of **Cd-Tmca-bpy** exhibit distinct characteristics when dispersed in different solvents such as ethanol, acetonitrile, N,N-Dimethylacetamide (DMAC), N,N-Dimethylformamide (DMF), methanol and water.

**Figure 10 polymers-16-02643-f010:**
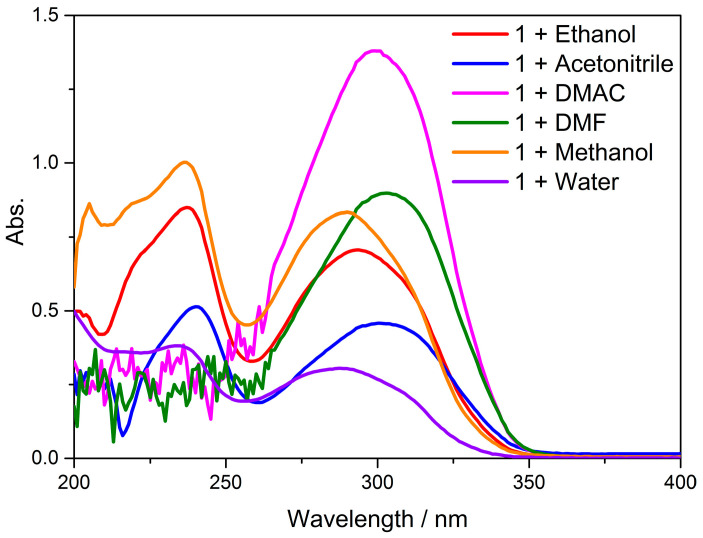
The UV–Vis absorption spectra of **Cd-Tmca-bpy** when dispersed in different solvents such as ethanol, acetonitrile, N,N-Dimethylacetamide (DMAC), N,N-Dimethylformamide (DMF), methanol and water.

**Figure 11 polymers-16-02643-f011:**
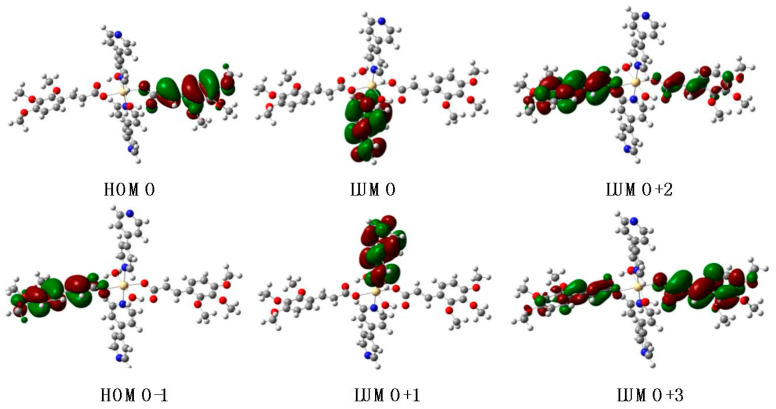
HOMO and LUMO orbitals in the optimized structures (B3LYP/6-31G(d)) for **Cd-Tmca-bpy**.

**Table 1 polymers-16-02643-t001:** Crystal data and structure refinement for **Cd-Tmca-bpy**.

	Cd-Tmca-Bpy
Empirical formula	C_17_H_19_Cd_0.5_NO_6_
Formula weight	389.53
Temperature/K	293(2)
Crystal system	monoclinic
Space group	C2/c
a/Å	28.753(12)
b/Å	11.848(5)
c/Å	10.632(5)
α/°	90
β/°	105.513(13)
γ/°	90
Volume/Å^3^	3490(3)
Z	8
*ρ*_calc_g/cm^3^	1.483
μ/mm^−1^	5.623
F(000)	1600.0
Crystal size/mm^3^	0.18 × 0.18 × 0.17
Radiation	CuKα(λ = 1.5478)
2Θ range for data collection/°	12.516 to 177.626
Index ranges	−37 ≤ h ≤ 37, −15 ≤ k ≤ 15, −13 ≤ l ≤ 13
Reflections collected	35,204
Independent reflections	3936 [*R*_int_ = 0.0243, *R*_sigma_ = 0.0130]
Data/restraints/parameters	3936/0/228
Goodness-of-fit on F_2_	1.146
Final R indexes [*I* >= 2σ(*I*)]	*R*_1_ = 0.0257, *wR*_2_ = 0.0674
Final R indexes [all data]	*R*_1_ = 0.0280, *wR*_2_ = 0.0693
Largest diff. peak/hole/e Å^−3^	0.76/−0.44

**Table 2 polymers-16-02643-t002:** Bond lengths for **Cd-Tmca-bpy**.

Bond	Length/Å	Bond	Length/Å
Cd1-O1 ^1^	2.2967(16)	Cd1-N1	2.375(3)
Cd1-O1	2.2968(16)	Cd1-N2 ^2^	2.400(3)
Cd1-O6	2.2958(17)	Cd1-O6 ^1^	2.2957(17)

^1^ 1 − X, +Y, 1/2 − Z; ^2^ +X, −1 + Y, +Z.

**Table 3 polymers-16-02643-t003:** Bond angles for **Cd-Tmca-bpy**.

Atoms	Angle/°	Atoms	Angle/°
O1^1^-Cd1-O1	179.61(8)	O6^1^-Cd1-O6	172.68(8)
O1^1^-Cd1-N1	90.19(4)	O1-Cd1-N1	90.20(4)
O1-Cd1-N2 ^2^	89.80(4)	O1^1^-Cd1-N2^2^	89.81(4)
O6^1^-Cd1-O1 ^1^	89.90(6)	O6-Cd1-O1^1^	90.07(6)
O6-Cd1-O1	89.90(6)	O6^1^-Cd1-O1	90.07(6)
O6-Cd1-N1	93.66(4)	O6^1^-Cd1-N1	93.66(4)
O6-Cd1-N2^2^	86.34(4)	O6^1^-Cd1-N2^2^	86.34(4)
N1-Cd1-N2^2^	180.0		

^1^ 1 − X, +Y, 1/2 − Z; ^2^ +X, −1 + Y, +Z.

**Table 4 polymers-16-02643-t004:** Hydrogen bonds for **Cd-Tmca-bpy**.

Hydrogen Bonds	Length/Å	Angle/°
O6-H6A-O1 ^4^	1.946	148.54
O6-H6B-O2	1.844	147.88
C10-H10A-O2 ^5^	2.538	152.16
C11-H11B-O5	2.491	114.68
C13-H13-O1 ^6^	2.621	170.82

^4^ +X, 1 − Y, −1/2 + Z; ^5^ −X + 1/2, −Y + 1/2, −Z; ^6^ −X + 1, −Y + 1, −Z + 1.

**Table 5 polymers-16-02643-t005:** Zone of inhibition (mm) of **Cd-Tmca-bpy** towards test bacteria in well-diffusion assays.

Complex	Cd-Tmca-bpy
*Staphylococcus epidermidis*	15
*Bacillus subtilis subsp. Spizizenii*	13
*Escherichia coli*	26
*Staphylococcus aureus*	22 *
*Pseudomonas aeruginosa*	18 *

* inhibition halo was hazy/indistinct.

## Data Availability

Data are contained within the article.

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
