# Peer review of "Crystal Structure, Photophysical Properties and Antibacterial Activity of a Cd(II) Complex with Trans-2,3,4-Trimethoxycinnamic Acid and 4,4′-Bipyridine Ligands"

_polymers, 2024, doi:10.3390/polym16182643_

Round 1

Reviewer 1 Report

Comments and Suggestions for Authors

This work presents the synthesis of a coordination polymer based on bipyridine, carboxylate and cadmium ions. The crystal structure is investigated and some properties, among which fluorescence, are studied. The description of the structure is quite acceptable. The assumption of electronic transitions to explain the fluorescence of the ligands is also not questionable.

Some questions are:

1. What can explain the difference in elemental analysis. Carbon is significantly overestimated compared to the expected value. Could this be due to the inclusion of additional solvent molecules (what kind and in what amount)?

2. “bacteriostatic activity” in the title of the article sounds strange. Replace it with a more common term.

3. What is the activity of the resulting polymer related to. Are there any assumptions based on literature data?

4. Can the fluorescence of a coordination polymer be characterized using lifetime or quantum yield?

5. Comparison of the polymer with known organometallic compounds should be made when discussing bacterial activity. Does carboxylate of cadmium without bipyridine or bipyridine with cadmium without carboxylate affect the selected bacteria? 

Author Response

Comments 1:

Comments and Suggestions for Authors

This work presents the synthesis of a coordination polymer based on bipyridine, carboxylate and cadmium ions. The crystal structure is investigated and some properties, among which fluorescence, are studied. The description of the structure is quite acceptable. The assumption of electronic transitions to explain the fluorescence of the ligands is also not questionable.

Some questions are:

1. What can explain the difference in elemental analysis. Carbon is significantly overestimated compared to the expected value. Could this be due to the inclusion of additional solvent molecules (what kind and in what amount)?

2. “bacteriostatic activity” in the title of the article sounds strange. Replace it with a more common term.

3. What is the activity of the resulting polymer related to. Are there any assumptions based on literature data?

4. Can the fluorescence of a coordination polymer be characterized using lifetime or quantum yield?

5. Comparison of the polymer with known organometallic compounds should be made when discussing bacterial activity. Does carboxylate of cadmium without bipyridine or bipyridine with cadmium without carboxylate affect the selected bacteria?

Response 1: Thank you for pointing this out. We agree with this comment. Therefore, we have made corresponding adjustments and improvements.

1. We have remeasured the elemental analysis of the complex. The result is as follows: C, 49.40; H, 4.65; N, 3.43.

Reasons for the differences between the measured values and theoretical values of elemental analysis: (1) There may be free water molecules in the complex crystal and it cannot be determined during the crystal structure analysis. By comparing the calculated values with the measured values, we speculate that there may be free water molecules in the composite crystal. (2) In theory, the final residual CdO in thermogravimetric analysis should be 16.48%, but the experimental value of the final residual amount is 17.08%.By combining thermogravimetric analysis with elemental analysis, it is speculated that there may be a small amount of cadmium compound mixed in the sample. This leads to a residual amount higher than the theoretical value of CdO produced only by the thermal decomposition of the complex.

2. The term "bacteriostatic activity" in the title of the article has been replaced by "antibacterial activity" to be consistent with the wording in the main text.

3. There are reports in the literature that complexes based on cinnamic acid derivatives have certain antibacterial effects. For example: [1] Shao-Song Qian, Yue-Hu Chen, Qi-Peng Long, Fang Wang, Hai-Liang Zhu. Syntheses, crystal structures, and antimicrobial activities of nickel(II) and cadmium(II) complexes with 4-methylsulfonyl cinnamate and diamines[J]. Journal of Coordination Chemistry, 2012, 65(24): 4419-4429. [2] Aragón-Muriel Alberto, Polo-Cerón Dorian. Synthesis, characterization, thermal behavior, and antifungal activity of La(III) complexes with cinnamates and 4-methoxyphenylacetate[J]. Journal of Rare Earths, 2013, 31(11): 1106-1113. We speculate that the complex {[Cd(C₁₂H₁₃O₅)₂(4,4'-bpy)(H₂O)₂]}n (CD-TMA-BPY), which is constructed with trans-2,3,4-trimethoxycinnamic acid (HTmca) and 4,4'-bipyridine (4,4'-bpy) as ligands, has antibacterial activity.

4. The absolute quantum yield of the complex Cd-Tmca-bpy were only 1.1% determined by using an FLS-980 fluorescence spectrometer. Due to the low quantum yield, we are difficult to measure the fluorescence lifetime of the complex.

5. We further determined the antimicrobial properties of (CH₃COO)₂Cd, HTmca, and 4,4'-bipyridine (4,4'-bpy) against Staphylococcus epidermidis (14990), Bacillus subtilis subsp. spizizenii (6633), Escherichia coli (25922), Staphylococcus aureus (12600), and Pseudomonas aeruginosa (15692). The results indicated that their antibacterial effects were inferior to that of the complex.

Reviewer 2 Report

Comments and Suggestions for Authors

the article " Crystal structure, photophysical properties and bacteriostatic activity of Cd(II) complex with trans2,3,4-Trimethoxycinnamic acid and 4,4'-Bipyridine ligands" show a good results in polymeric structure by reaction of two different monomers and prove the structure with different methods but the article needs minor revision as follow

1- the abstract is too long compared with other parts of the paper

2- where is the CCDC for the  single crystal 

3- the presence of carbonyl group in IR must be re-investigated 

Author Response

Comments 2:

Comments and Suggestions for Authors

the article " Crystal structure, photophysical properties and bacteriostatic activity of Cd(II) complex with trans2,3,4-Trimethoxycinnamic acid and 4,4'-Bipyridine ligands" show a good results in polymeric structure by reaction of two different monomers and prove the structure with different methods but the article needs minor revision as follow

1- the abstract is too long compared with other parts of the paper

2- where is the CCDC for the single crystal

3- the presence of carbonyl group in IR must be re-investigated

Response 2: Agree. We have, accordingly, modified abstract, Single-crystal structure determination, IR spectrum to emphasize this point.

1. The abstract section has been meticulously revised. We carefully extracted the crucial content and have now presented it in a clear and prominent manner.

2. We have added the CCDC number to the section titled “2.2 Single-crystal structure determination”(page 4, line 45). Additionally, we have uploaded the cif file and check cif file as supporting information to enhance the completeness of the data.

3. The presence of the carbonyl group in the IR spectrum has been thoroughly reinvestigated. Moreover, a comprehensive list of functional groups as presented in Fig. 6 within the supplementary file is provided for a more detailed analysis.(page 9-10)

Reviewer 3 Report

Comments and Suggestions for Authors

1. The word "difffferent" was spelled incorrectly. Please revise. I also recommend authors to check all grammar in the manuscript to make sure they are all correct.

2. I highly recommend authors to change "1" to other code name that better represent your complex.

3. Can authors list names of microbials used for the antimicrobial ability in the methodology section?

4. Can authors provide a list of functional groups presented in Fig. 6, probably in a supplementary file?

5. Can authors add the derivative weight loss to the TGA result? Any related discussion should be added as well.

6. In section 3.8, authors said that 5 gram-positive and 4 gram-negative bacteria were used for the test, however, only 5 were shown in Table 5.

Comments on the Quality of English Language

Minor check is needed.

Author Response

Comments 3:

Comments and Suggestions for Authors

1. The word "difffferent" was spelled incorrectly. Please revise. I also recommend authors to check all grammar in the manuscript to make sure they are all correct.

2. I highly recommend authors to change "1" to other code name that better represent your complex.

3. Can authors list names of microbials used for the antimicrobial ability in the methodology section?

4. Can authors provide a list of functional groups presented in Fig. 6, probably in a supplementary file?

5. Can authors add the derivative weight loss to the TGA result? Any related discussion should be added as well.

6. In section 3.8, authors said that 5 gram-positive and 4 gram-negative bacteria were used for the test, however, only 5 were shown in Table 5.

Response 3: Thank you for pointing this out. We agree with this comment. We have made corresponding adjustments and improvements.

1. It has been carefully checked. During this process, all misspelled words have been meticulously identified and corrected to ensure the accuracy and integrity of the text.

2. The abbreviation of the complex was redefined from “1” to “Cd-Tmca-bpy”.

3. The names of the microorganisms used for assessing antimicrobial ability are listed in the methodology section. (page 6)

4. A list of functional groups presented in Fig. 6 in the supplementary file is provided.

5. We have added the derivative weight loss data to the result of the thermogravimetric analysis (TGA), enhancing the comprehensiveness of the analysis, and updated Figure 7 accordingly.(page 10)

6. The microorganisms with obvious antibacterial effects are listed in Table 5. Others have been added in the text.(page 14)

Round 2

Reviewer 3 Report

Comments and Suggestions for Authors

Authors have revised the manuscript and answered to my comments very well. The manuscript is now ready for publication.

Comments on the Quality of English Language

Minor check is needed.